# Mechanism and Influence of Dispersants on the Action of Polymer Flocculants Used in Slurry Separation

**DOI:** 10.3390/polym15204073

**Published:** 2023-10-12

**Authors:** Guoping Ren, Jian Zhang, Tugen Feng, Yu Liang, Yihao Yin

**Affiliations:** 1Key Laboratory of Ministry of Education for Geomechanics and Embankment Engineering, Hohai University, No. 1 Xikang Road, Nanjing 210024, China; 18355267268@163.com (G.R.); fengtugen@hhu.edu.cn (T.F.); 2Shenzhen Campus, Sun Yat-sen University, No. 66 Gongchang Road, Shenzhen 518107, China; liangyu25@mail.sysu.edu.cn; 3School of Civil Engineering, Sun Yat-sen University, No. 2 Daxue Road, Zhuhai 528478, China; 4China Railway 14th Bureau Group Da Shield Engineering Company Limited, Nanjing 211800, China; 13865200407@163.com

**Keywords:** slurry separation, dewatering, flocculant, dispersant, mechanism

## Abstract

The application of polymer flocculants plays a pivotal role in the slurry separation process of shields, and the dispersant used for treating cutter mud cakes can significantly impact the effectiveness of polymer flocculants, potentially leading to reduced efficiency in slurry separation. Experiments were conducted to select appropriate flocculants and investigate the influence of dispersants on flocculant effectiveness, aiming to assess the effect of flocculants and explore the relationships and mechanisms governing their influence. Changes in the patterns of slurry flocculation were revealed in terms of flocculation-driven precipitation and vacuum-filtration effects. The purpose of this article is to provide a reference for the field application of polymer flocculants in the shield field. The conclusions are as follows. Inorganic flocculants containing 0.5% polyaluminum chloride (PAC) exhibit the most effective flocculation, demonstrating strong charge neutralization action. Organic flocculants containing 0.1% cationic polyacrylamides (CPAM) exhibit the most effective flocculation, demonstrating strong bridging and net capture effects. The dispersant sodium hexametaphosphate (SHMP) can significantly weaken the charge-neutralizing action of flocculants and slightly enhance bridging and net capture effects. SHMP can impede the flocculation of slurry with PAC. For CPAM, SHMP can enhance the flocculation of slurry at a low mass fraction (0.1% and 0.3%), while SHMP can significantly hinder flocculation at a high mass fraction (0.5% and 1%). A low mass fraction of SHMP reduced slurry viscosity to 246.3 mPa.s and enhanced vacuum filtration, while a high mass fraction of SHMP increased slurry viscosity to 667.2 mPa.s and hindered vacuum filtration. In conclusion, while dispersants reduce the effectiveness of inorganic flocculants at any mass fraction, a small number of dispersants enhances the performance of organic flocculants; thus, the organic flocculant CPAM is recommended for slurry separation.

## 1. Introduction

Slurry shields effectively regulate soil and water pressure on excavation faces by injecting slurry into sealed soil bins, enabling the precise and rapid adjustment of the working pressure to ensure stability at working faces. This technology has extensive applications in large-diameter tunnel construction projects. The gravity and viscosity of shield slurries progressively increase during extended driving processes, giving rise to various technical challenges, such as a reduction in shield driving rate, an elevation in cutter torque, and an augmented risk of cutter mud caking. Therefore, to ensure the stability of shield excavation faces, protect the environment along the lines, and facilitate smooth shield excavation, it is necessary to separate mud from water and control the slurry index value [1].

Shield slurry separation technology is a complex process that involves several steps, such as vibration screening [2], multistage swirling [3], vacuum filtration [4], and centrifugation [5]. The aforementioned slurry separation methods encompass a range of techniques, yet the pivotal stage in this process lies in the introduction of flocculants. Flocculants have the capacity to aggregate, settle, and eliminate colloids and particles within the slurry, thereby establishing favorable conditions for subsequent stages of slurry separation [6]. Polymer flocculants stand out as one of the most prevalent reagents for flocculation. The application efficiency of flocculants directly impacts the effectiveness of slurry separation; therefore, scholars have conducted extensive research to investigate the factors that influence the efficiency of flocculants. Studies conducted by He et al. [7], Tripathy and Singh [8], Min et al. [9], and Tsubaki et al. [10] explored the impact of different types of flocculants on the efficiency of slurry separation. For instance, organic flocculants exhibit faster dewatering time; however, the resulting separated sludge included a significant amount of water, thereby impeding its transportation [7,8]. Although the inorganic flocculants of the iron and aluminum series exhibit favorable dewatering efficacy, they are associated with prolonged dewatering duration and pose environmental pollution concerns for the separated wastewater [9]. Although the novel microbial flocculant exhibits remarkable dewatering effectiveness, its exorbitant cost renders it unsuitable for large-scale field application [10]. Wang et al. [11] compared and analyzed the effects of a variety of traditional inorganic and organic flocculants on slurry separation efficiency and found that both inorganic and organic flocculants could improve slurry separation efficiency, but the effect of the inorganic flocculant ferric chloride was not as good as that of the organic flocculant PAM. To improve the settling performance of red mud dissolved from imported bauxite, Liang et al. [12] prepared a new oxyamic acid-type flocculant. After slurry separation, the clarification and settling speed of the supernatant were better than those of traditional industrial flocculants. Kurade et al. [13], Guan et al. [14], Taylor et al. [15], and Romero et al. [16] investigated the impact of test conditions on slurry separation efficiency. Allaedini and Zhang [17] posited that temperature plays a crucial role in influencing the efficacy of flocculants. To investigate this, precipitation tests were conducted at temperatures of 50, 60, 75, and 100 °C. Notably, it was observed that at a temperature of 60 °C, the flocculants exhibited superior performance for slurry separation through sedimentation. The effects of various conditions, such as pH and stirring rate, on the flocculation of lignite waste were investigated by Sarioglu et al. [18]. Experimental data revealed that a cationic flocculant exhibited minimal sensitivity to changes in pH, whereas an anionic flocculant was highly influenced by variations in pH.

In addition, a slurry shield is equipped with a slurry circulation system to ensure the stability of excavation faces. The addition of an admixture at the excavation face significantly impacts the efficacy of flocculants in this circulation system. Dispersants are commonly used additives in slurry separation, primarily employed to prevent and treat the formation of cutter mud cakes during shield tunneling processes [19,20]. During the shield machine’s excavation process, a water gun positioned at the end of the soil bin typically injects a dispersant at a mass fraction ranging from 0.01% to 2% into both the soil bin and cutter head [21]. The introduced dispersant molecules gradually permeate through the mud cake, reducing the adhesive forces between the clay and the shield cutter as well as other machinery components. This approach effectively addresses issues related to cutter mud cake formation [22]. Liu et al. [23] demonstrated that the application of dispersants, such as sodium polyacrylate, effectively enhances the absolute zeta potential of soil particles, thereby increasing the maximum repulsive energy between particles and subsequently weakening interparticle connections in shield soils. This reduction in connectivity mitigates the risk of mud cake formation. Zhao et al. [24] investigated the impact of an anionic sodium-salt copolymer dispersant on shield slurries and mud cakes at varying concentrations. Their results revealed that the dispersant effectively reduced both the empirical viscosity ratio of the slurry and the suspension viscosity, thereby mitigating mud cake formation. Notably, while the aforementioned scholars observed a positive impact of dispersants on cutter mud cake treatment, Wang et al. [25] and Pagnoli et al. [26] demonstrated that dispersants carrying negative charges counteract the mechanisms active in flocculants. The dispersant remaining in the soil bin at the front end of a shield tunneling machine will be introduced into the slurry separation system on the ground through the slurry circulation system, further impacting the efficacy of flocculants in subsequent slurry separation processes. Hence, it is imperative to investigate the influence of dispersants on slurry flocculation for shield slurry separation.

Several scholars have also conducted research on the impact of dispersants on the efficacy of flocculants; however, limited literature is available for reference. For instance, Ma [27] investigated the impact of different addition sequences of dispersant and flocculant starch on the selective flocculation of hematite, revealing that starch and the dispersant exhibited contrasting effects on zeta potential. Moreover, pretreatment of hematite with dispersant was found to be unfavorable for effective flocculation and precipitation. Rodrigo et al. [28] investigated the impact of the dispersant sodium phosphatic acid on the flotation efficiency of molybdenum ore in seawater and observed that the influence of SHMP on flocculant polyacrylamide was not entirely detrimental. Despite an increase in supernatant turbidity following SHMP usage, smaller and denser aggregates formed, which accelerated the settling rate. Tomokazu et al. [29] investigated the impact of polygamma-glutamic acid flocculant (PGAF) on the removal efficiency of ultrafine cement (UFC) particles stabilized by polycarboxylic acid copolymers, known as high-efficiency water-reducing agents (SP). Their findings revealed that PGAF exhibited promising potential for effectively flocculating and settling pure cement removal performance was observed under conditions characterized by low ionic strength and efficient rapid mixing. Although the aforementioned scholars have conducted comprehensive research on the efficacy of dispersants in mud cake treatment, the majority of their studies relate to chemical and mining engineering [30,31], with limited coverage of the slurry conditions of slurry shields. Furthermore, there is a lack of literature elucidating the mechanism by which dispersants influence flocculants, with most scholars primarily analyzing causal factors through charge theory alone [25,27]. Therefore, it is necessary to conduct more comprehensive and in-depth quantitative investigations to bridge existing gaps in this field.

In light of the aforementioned issues, an optimization test on flocculants was initially conducted in this study for the Zhanjiang Bay undersea Tunnel in Guangdong Province. A selection was made based on flocculation–precipitation tests among various commonly used on-site polymer flocculants, resulting in the identification of an organic and an inorganic flocculant. Subsequently, the impact of dispersants on various types of polymer flocculants was investigated by conducting tests to assess their influence on flocculation efficacy. Finally, the physical and chemical effects that underlie the differences between dispersants and inorganic/organic flocculants were elucidated. As the slurry separation efficiency of shield directly impacts tunneling productivity and construction safety, this paper aims to consolidate the influence of dispersants on flocculants and propose a set of practical application schemes for slurry separation reagents with engineering significance, aiming to inspire and provide suggestions for researchers in related fields regarding the utilization of flocculants.

## 2. Project Overview

The Zhanjiang Bay Undersea Tunnel in Guangdong Province is a comprehensive control project for the Guangzhou–Zhanzhan high-speed railway, representing the longest large-diameter cross-sea high-speed railway tunnel with a single head that is currently available, spanning an impressive total length of 7352 m. The water section spans a length of 2.5 km, while the shield tunnel boasts a diameter of 14.3 m. During its operation, the shield encountered various geological strata, including Quaternary Holocene silty clay, clay, medium sand, and coarse sand. An overview of the project is presented in Figure 1. Based on geological exploration data, it was observed that the formation predominantly consisted of silty clay or clay particles of sizes less than 50 μm, accounting for over 60% of the composition. The overall uniformity of the formation was found to be poor, leading to frequent occurrences of mud cake deposition on the cutter head. Consequently, dispersants are regularly employed for the on-site treatment of cutter mud cakes.

The slurry treatment system for the Zhanjiang Bay Undersea Tunnel is situated above ground level and comprises four components: a slurry circulation system, a slurry separation system, a slurry preparation and mixing system, and a slurry monitoring system. The slurry separation system includes vibrating screens, first- and second-stage cyclones, sedimentation tanks, and centrifugation equipment. The separation efficiency of field cyclones and centrifugation is unsatisfactory, as the waste pulp exhibits high gravity viscosity and poor sedimentation even after treatment. Therefore, prior to slurry separation, it is imperative to incorporate suitable flocculants into the slurry. Figure 2 illustrates the specific flocculant employed for slurry separation, along with the resultant wastewater and sludge properties after this process.

## 3. Materials and Methods

### 3.1. Experimental Materials

The slurry collected at the Zhanjiang site and its particle distribution curve are presented in Figure 3. The particle distribution of the mud was determined by conducting three wet measurements using a Mastersizer 2000 laser particle size meter (measuring range from 0.1 μm to 3000 μm). The slurry exhibits a relatively high proportion of fine particles, with approximately 75% of the total particles having particle sizes less than 10 μm. In the process of slurry transportation, water loss occurs due to evaporation and other factors. Therefore, to simulate field conditions and facilitate observation, the volume of collected waste slurry was diluted by a factor of three. The specific gravity of the collected waste slurry was determined to be approximately 1.113 g/cm^3^, with a solid content of approximately 13.4%.

To evaluate the impact of dispersants on the efficacy of polymer flocculants, reagents commonly employed in the respective fields were selected for this experiment. The flocculants under investigation are categorized as inorganic and organic. In the inorganic category, there is polyferric chloride, polyferric sulfate, and polyaluminum chloride, while in the organic category, there is anionic polyacrylamide and cationic polyacrylamide. For mud cake treatment at the Zhanjiang site, sodium hexametaphosphate was employed as a dispersant. Its properties can be found in Table 1.

### 3.2. Experimental Methods

First, an organic flocculant and an inorganic flocculant were selected through a flocculation–precipitation test. Subsequently, the impact of dispersants on different types of flocculants was investigated by assessing their influence on the effectiveness of the flocculation process. Finally, the dissimilarities between dispersant and inorganic/organic flocculants were analyzed and elucidated from both physical and chemical perspectives.

#### 3.2.1. Flocculant Optimization Test

In this experiment, a flocculation–sedimentation test was employed. First, a screening test was conducted to determine the optimal dose of the five types of flocculants. The turbidity of the supernatant was measured using a knob-type turbidimeter (model WGZ-1B) and the slurry separation volume ratio obtained from a measuring cylinder was utilized as evaluation indices for determining the optimum dose of each flocculant (addition amount = reagent dry weight/solid dry weight in slurry). The five types of flocculants were tested sequentially as follows: a specific amount of slurry was placed in individual 500 mL beakers, and subsequently, a solution of each flocculant at varying concentrations was added to each beaker containing 50 mL distilled water. The concentration levels for each reagent were determined based on preliminary experiments and engineering practices, which are detailed in Section 4.1.1, as presented in the table below. The slurry was mixed with 50 mL of flocculant solution and vigorously stirred at a speed of 100 r/min for 1 min. After allowing the mixture to precipitate for 1 h, the turbidity of the supernatant and the volume ratio of slurry separation (volume ratio = supernatant liquid volume/slurry total volume) were measured.

The optimal type of flocculant was determined based on the optimum dose of each flocculant. Following a testing method similar to that described above, the size of alum flowers was observed, and the turbidity of the supernatant along with the volume of separated supernatant liquid from the slurry were recorded.

#### 3.2.2. Test on the Effect of Dispersants on Flocculation

According to the aforementioned tests, an inorganic flocculant and an organic flocculant were chosen. Subsequently, varying mass fractions of dispersants were separately added to investigate their impact on the flocculation and vacuum-filtration efficacy of the slurry. The test to determine flocculation was similar to that described above, while the vacuum-filtration test had additional steps. The simplified dewatering apparatus depicted in Figure 4 was employed to simulate the mechanical dewatering process in Zhanjiang. After slurry separation, a 300 mL sample was extracted from the lower part of the measuring cylinder and thoroughly mixed with reagents before being poured into the glass container positioned above the dewatering device. Subsequently, filter paper was placed in the lower section, and a filter extraction machine was connected to facilitate vacuum filtration and dewatering of the treated slurry. The water content of the sludge was measured after 60 min.

#### 3.2.3. Test of the Influence Mechanism of Dispersant

In this experiment, scanning electron microscopy (SEM), particle size analysis, and zeta potential were employed to further investigate the reasons and mechanisms underlying the disparities between the dispersant and flocculant in terms of their impacts on slurry flocculation and vacuum filtration. Table 2 presents the test parameters, instruments utilized for mechanism testing, and the significance of the analyses.

## 4. Results and Discussion

### 4.1. Flocculant Optimization Test

#### 4.1.1. Screening for Optimal Addition Amount

The optimal dose of five types of flocculants was determined by conducting flocculation and precipitation tests. The test results obtained after 1 h are presented in Table 3, and Figure 5, Figure 6, Figure 7, Figure 8 and Figure 9 illustrate diagrams depicting the flocculation effect along with corresponding changes in turbidity and the volume ratio for each group.

The slurry obtained after the PFC action is depicted in Figure 5. Initially, the slurry separation volume ratio increased and then decreased with increasing flocculant dose. The maximum volume ratio of 22.12% was achieved at an addition of 1%. At an addition of 0.5%, the supernatant exhibited a turbidity peak of 43 NTU, while the minimum turbidity of 18 NTU was observed at an addition of 1%. Considering both indices, the optimal addition of PFC to the slurry was determined to be 1%.

The slurry obtained after PFS action is depicted in Figure 6. The supernatant volume change during slurry separation exhibited poor performance, displaying an initial increase followed by a subsequent decrease. A maximum volume ratio of 10.91% was achieved at an addition of 0.5%. The supernatant in each group appeared relatively clear, and the turbidity did not exhibit significant changes with increasing amounts of PFS added. The lowest turbidity value for the supernatant was observed at an addition of 1%–2%, averaging 15 NTU. Considering both indices, the optimal addition of PAC to the slurry was determined to be 1%.

The slurry obtained after PAC treatment is depicted in Figure 7. The slurry separation effect was evident for all groups, except for the group with 2% PAC added. The volume ratio of slurry separation initially increased and then decreased with increasing amount of PAC added. The maximum volume ratio observed was 32.26% at an addition of 0.5%. Except in the 0.5% group, each group exhibited a clear supernatant. Among all groups, the turbidity of the supernatant was lowest at 0.5%, at only 16 NTU. Considering both indices, it can be concluded that the optimal addition of PAC to the slurry was 0.5%.

The slurry obtained after the action of APAM is depicted in Figure 8. The slurry separation effect was pronounced across all groups, with the volume ratio of slurry separation initially increasing and then decreasing as the additive amount increased. The maximum volume ratio reached 32.73% at an addition of 0.1%. However, it should be noted that the turbidity of each group’s slurry supernatant exhibited a similar trend, initially increasing and then decreasing. Notably, when the addition ranged from 0.1% to 0.5%, the turbidity was low, with an average value of 86 NTU. Considering both indices collectively, it can be concluded that the optimal addition of APAM to the slurry was 0.1%.

The slurry obtained after CPAM treatment is depicted in Figure 9. The slurry separation effect was pronounced across all groups, with a decreasing trend in the volume ratio of separated slurry as the amount of additive increased. The maximum volume ratio reached 43.18% at an addition of 0.1%. The supernatant of the slurry exhibited an initial increase followed by a decrease but remained turbid in all groups, with an average turbidity level of 79 NTU. Considering both indices, the optimal addition of CPAM to the slurry was determined to be 0.1%.

#### 4.1.2. Screening of Optimal Species

The flocculation and precipitation effects of five types of flocculants were tested at their optimal mass fractions: PFC (1%), PFS (1%), PAC (0.5%), APAM (0.1%), and CPAM (0.1%). Figure 10 illustrates the comparison among these groups. In terms of turbidity reduction in the separated supernatant, the order is as follows: PAC > PFC > PFS > CPAM > APAM. Consequently, clearer supernatants are generally obtained with the application of inorganic flocculants to the slurry. The volume of separated supernatant liquid follows this sequence: PFS < PFC < PAC < APAM < CPAM. Hence, organic flocculants combined with slurry typically result in larger volumes of separated supernatant liquid. Among the inorganic flocculants, PAC exhibits superior flocculation and precipitation effects, separating 29.70% of the supernatant within 60 min, at a turbidity level of 15 NTU. Among the organic flocculants, CPAM exhibits superior flocculation and precipitation effects, separating 45.52% of the supernatant within 60 min, at a turbidity level of 74 NTU.

### 4.2. Test on the Effect of Dispersants on Flocculation

#### 4.2.1. Effect of Dispersants on Inorganic Flocculants

Figure 11a illustrates the change observed in supernatant liquid volume after slurry separation following the addition of the dispersant SHMP and the inorganic flocculant PAC. The trend in SHMP groups at concentrations of 0.1%, 0.3%, and 0.5% exhibited similarity to the group without dispersant addition, while a change in supernatant liquid volume was not obvious over 40 min. A change in the liquid volume of the supernatant in the 1% group is not significantly discernible, exhibiting slow change and a nearly linear trend. Within 60 min, 113.2 mL, 97.2 mL, 59.3 mL, and 18.1 mL of supernatant were separated in the SHMP groups with concentrations of 0.1%, 0.3%, 0.5%, and 1%, respectively, which was significantly smaller than separation without the addition of a dispersant (178.6 mL). Therefore, any concentration of dispersant has a certain inhibitory effect on slurry flocculation. Figure 11b illustrates the temporal trend in supernatant turbidity after slurry separation. The turbidity of the 0.1% and 0.3% groups is marginally higher than that of the group without dispersant addition, suggesting that low concentrations of dispersant (0.1% and 0.3%) exhibit a modest inhibitory effect on supernatant turbidity changes. The 0.5% and 1% groups showed values that remained in the high turbidity range, at approximately 97 NTU and 123 NTU, respectively, demonstrating a pronounced inhibitory effect of the concentrated dispersants (0.5% and 1%) on slurry turbidity changes.

The water content of sludge, as measured in the slurry after vacuum filtration, is illustrated in Figure 12. The addition of the dispersant SHMP at concentrations of 0.1% and 0.3% resulted in a certain degree of enhancement in the vacuum-filtration efficiency of the slurry. Decreased water content facilitates convenient on-site drying and transport for subsequent treatment. The addition of the dispersant SHMP at concentrations of 0.5% and 1% significantly impedes the vacuum-filtration process for slurry, leading to elevated sludge water contents of 43.5%, 49.2%, and 55.1%. The results show that a low concentration of dispersant can effectively decrease the water content of sludge, while a high concentration of dispersant has the potential to increase it.

#### 4.2.2. Effect of Dispersants on Organic Flocculants

Figure 13a illustrates the temporal change in supernatant liquid volume after slurry separation following the addition of the dispersant SHMP and the organic flocculant CPAM. The separation was 373.5 mL and 353.2 mL in the SHMP groups with concentrations of 0.1% and 0.3%, respectively, exhibiting significantly higher separation than without a dispersant, at 325.6 mL. Hence, it can be inferred that the addition of a low concentration of dispersant (0.1% and 0.3%) effectively enhances slurry flocculation. The separation was 243.8 mL and 197.2 mL in the SHMP groups with concentrations of 0.5% and 1%, respectively, exhibiting a significantly lower separation than without a dispersant, at 325.6 mL. Hence, it can be inferred that the addition of a high concentration of dispersant (0.5% and 1%) effectively inhibits slurry flocculation. Figure 13b illustrates the temporal change in supernatant turbidity after slurry separation. Within 60 min, the SHMP group achieved supernatants with turbidity levels of 50 NTU and 77 NTU at concentrations of 0.1% and 0.3%, respectively, demonstrating improved clarity compared to the control group without dispersant addition (99 NTU). Consequently, the inclusion of low concentrations of dispersant (0.1% and 0.3%) had a discernible effect in reducing supernatant turbidity. The supernatant turbidity in the 0.5% and 1% SHMP groups, with values of 154 NTU and 250 NTU, respectively, exhibited higher levels compared to the dispersant-free sample, which had a value of 99.6 NTU. Notably, the addition of 1% dispersant completely impeded the flocculating effect of the flocculant. Consequently, high concentrations of dispersants (0.5% and 1%) significantly hindered alterations in supernatant turbidity.

The water content of sludge, as measured after vacuum filtration, is illustrated in Figure 14. The addition of the dispersant SHMP at concentrations of 0.1%, 0.3%, and 0.5% results in some degree of enhancement in the vacuum-filtration efficiency of the slurry. Specifically, the water content in the separated slurry is reduced by 9%, 4.3%, and 1.6%, with the most significant improvement observed at a concentration of 0.1% dispersant. The addition of 1% SHMP dispersant partially inhibits the vacuum-filtration process of the slurry, resulting in an increase in the water content of the separated sludge to 47.2%. Therefore, low concentrations of dispersants decrease the sludge water content, while high concentrations of dispersants increase it.

### 4.3. Test of the Influence Mechanism of Dispersant

#### 4.3.1. The Flocculant Mechanism

Prior to assessing the impact of dispersants on flocculation, it is imperative to comprehend the mechanism of action of the flocculant. According to the XDLVO theory, water and cations, as shown in Figure 15, form a double-layer structure with slurry particles. The double electric layer structure includes a compact layer and a diffusion layer. The compact layer is composed of negative ions and tightly adsorbed positive ions on the surface of slurry particles, while the diffusion layer is composed of positive ions moving freely in water. The forces between slurry particles include van der Waals force, steric resistance force, and electrostatic repulsion force [32]. The van der Waals gravitational force is the sum of dispersion force, polarity force, and induced dual force between particles; its magnitude depends on the distance between particles. The electrostatic repulsion force is generated by the interaction of the layer around the particle when it approaches; there exists a functional relationship between repulsion force and particle spacing index. Steric force occurs when particles are close enough to each other and significantly affect each other. The flocculation of slurry particles is influenced by these. After being dissolved in water, flocculants form chain structures that actively collide with slurry particles through Brownian motion [33]. Flocculant molecules adsorb onto slurry particle surfaces and alter various interaction forces among slurry particles to affect their flocculation state. 

The upper part of Figure 16 illustrates that upon dissolution in water, the polymer flocculant forms a high-quality and elongated molecular chain structure. This molecular chain possesses numerous charged active groups, which can be attracted to hydrogen bonds present on the slurry surface, facilitating the aggregation of small slurry particles into larger flocs. If it is a metal salt flocculant when introduced into water, metal salts such as aluminum and iron undergo hydrolysis to generate a substantial amount of hydrated metal oxide precipitates with a three-dimensional structure. As the volume of these hydrated metal oxides contracts and settles, colloidal particles and suspended particles in the water are effectively entrapped and removed akin to a filtration process. This action is also known as sweep coagulation. It leads to an enhanced aggregation of flocs, which facilitates their gravitational settling, ultimately achieving efficient separation. This intricate process includes bridging and net capture action, which are characterized by complex physical and chemical phenomena [34]. As depicted in the lower half of Figure 16, the slurry particles exhibit mutual repulsion due to the accumulation of identical charges on their surfaces. This electrostatic repulsion leads to a thermodynamically stable state in the slurry, rendering it resistant to precipitation. When dissolved in water, the flocculant generates ions with opposite charges that adsorb onto slurry particles through electrostatic attraction and neutralize surface charge. The compressive electrostatic force reduces the double electric layer of slurry particles, enhances the gravitational effect, and causes flocculation of the slurry particles. Simultaneously, through van der Waals forces, chemical bond forces, hydrogen bond forces, and bridging actions, the flocculant aggregates colloidal particles together to form, settle down, and facilitate precipitation of the slurry. This process is primarily governed by chemical principles and is referred to as charge neutralization action [35].

The utilization efficiency of flocculants is jointly influenced by the multiple aforementioned factors, rendering it a highly intricate process. While a flocculant may exhibit advantages from one perspective, it may also possess disadvantages from another. Consequently, the overall effectiveness of flocculation results from the synergistic effects of various actions, necessitating a comprehensive analysis of alterations in flocculation efficacy from diverse perspectives. Following scanning electron microscopy (SEM), particle size, zeta potential, and slurry viscosity analyses were employed to comprehensively analyze the causes and mechanisms underlying variations in the flocculation and vacuum-filtration effects of the dispersant SHMP after its interaction with the organic flocculant CPAM and inorganic flocculant PAC.

#### 4.3.2. Effect of Dispersant on Bridging and Net Capture Action

SEM micrographs (500 magnification) of the dried slurry are shown in Figure 17, illustrating the impact of various flocculants or dispersants used within each experimental group. The particle size distribution, as determined by the particle size meter, is depicted in Figure 18. The original slurry, as depicted in Figure 17a and Figure 18, exhibits a homogeneous structure, regular arrangement, moderate spacing, and predominantly small particles, with 78.12% of them measuring less than 50 μm. In this state, the slurry has stable properties and demonstrates resistance to precipitation. However, noticeable disparities are observed in the slurry following the introduction of the organic flocculant CPAM (Figure 17b) and inorganic flocculant PAC (Figure 17d). The slurry particles, under the influence of the organic flocculant CPAM, underwent substantial adsorptive connections and formed larger viscous particles with diameters exceeding 50 μm (43.69%). The particle surfaces were rough, while their distribution appeared to be random. The slurry treated with the inorganic flocculant PAC also exhibited a certain level of particle aggregation and decreases in interparticle distance; however, the degree of aggregation was significantly smaller than that observed in the slurry treated with CPAM (26.88% had a diameter larger than 50 μm). The slurry particles exhibit a smooth lamellar structure, and the particle distribution demonstrates a regular and loosely dislocated arrangement. The agglomeration of small particles represents one of the fundamental principles underlying slurry flocculation, specifically involving bridging and net capture mechanisms. After the addition of the dispersant SHMP to the slurry, as depicted in Figure 17c, the morphology of slurry particles remained unaltered following treatment with the organic flocculant CPAM; however, an emergence of particles with larger diameters was observed. In Figure 17e, both the particle morphology and size distribution of the slurry exhibited no changes after treatment with the inorganic flocculant PAC.

The particle size gradation diagram in Figure 18 provides a quantitative explanation for the same phenomenon. Upon the addition of equal mass fractions of flocculant and dispersant, the curves for all four sample types shift to the right relative to the original slurry. The shift occurs in the following order: CPAM + SHMP > CPAM > PAC ≈ PAC + SHMP > original slurry. As the shift increases, the reagent demonstrates enhanced bridging and net capture action. The proportion of large particles larger than 50 μm in the slurry after CPAM and PAC treatment was determined to be 43.69% and 26.88%, respectively. Consequently, it can be inferred that CPAM exhibited superior bridging and net capture action compared to PAC. Under the influence of gravity, flocs with larger particle sizes demonstrate enhanced precipitation efficacy. This phenomenon contributes to the superior flocculation and precipitation performance exhibited by the organic flocculant CPAM group in Section 4.1.2 compared to the inorganic flocculant PAC group. Compared to the original slurry, which accounted for 21.88%, the bridging and net capture effects only resulted in a mere 5.0% increase in size for particles 50 μm. Hence, it is plausible that the primary cause of PAC-induced flocculation was not only bridging and net capture. After the addition of the dispersant SHMP, the particle size distribution in the slurry with PAC remained essentially unaltered, indicating that SHMP had minimal impact on the bridging and net capture action of inorganic flocculants. The size distribution of large slurry particles exhibited a slight increase with CPAM, with an observed 5.88% increase in the proportion of particles larger than 50 μm. Consequently, the addition of the dispersant SHMP resulted in some degree of enhancement in the bridging and net capture action of the organic flocculant CPAM, accelerating the process of slurry flocculation and precipitation. The enhancement in the bridging and net capture action of the organic flocculant CPAM by SHMP is attributed to the unique molecular structure of CPAM. Upon dissolution in water, the elongation capacity of its long chain is inherently limited. However, the addition of SHMP facilitates the dissociation of CPAM, further elongating the lengthened chain and activating a greater number of adsorption sites for slurry particles. Moreover, it promotes both the bridging and net capture action of CPAM [36]. This is also one of the reasons for the difference between the effect of the dispersant and the inorganic and organic flocculants in Section 4.2.1 and Section 4.2.2.

#### 4.3.3. Effect of Dispersant on Charge Neutralization

The zeta potential of slurry particle changes that occur with the addition of a dispersant, as measured following the addition of two flocculants, are depicted in Figure 19. The zeta potential value also serves as an indicator of the stability of the dispersed system. A higher absolute value of the zeta potential corresponds to enhanced system stability, while a lower zeta potential suggests an increased likelihood of precipitation [37]. The initial zeta potential of the slurry was −30 mV, and upon the addition of an equivalent mass fraction of organic flocculant, the potentials of the slurry with CPAM and PAC increased to −21.9 mV and −12.6 mV, respectively. The stability of the slurry was disrupted by the charge of the flocculant, which corresponds to a mechanism known as charge neutralization. The decrease in the zeta potential of slurry particles after the addition of the organic flocculant CPAM is smaller than that observed with the inorganic flocculant PAC. Consequently, it can be inferred that charge neutralization by the inorganic flocculant PAC is stronger as it has a higher content of metal cations such as aluminum. As a result, at equivalent doses, the cation concentration of the inorganic flocculant PAC is higher.

After the addition of the dispersant SHMP, the zeta potential of both inorganic and organic flocculants exhibited some reduction. Particularly at an addition level of 1%, the zeta potential of slurry particles with inorganic and organic flocculants decreased to values even less than that of the original slurry, reaching −36.1 mV and −32.1 mV, respectively. Consequently, the stability of the slurry system was enhanced, rendering it more resistant to settling. The presence of the dispersant SHMP hinders flocculation from a charge perspective due to its ability to introduce additional anions. Moreover, the zeta potential curve of inorganic flocculants exhibits a linear decrease with increasing dispersant concentration, resulting in an average reduction of 2.3 mV in zeta potential for every 0.1% addition of SHMP. The zeta potential of the organic dispersant remained relatively stable at low concentrations (0.1% and 0.3%) but exhibited a rapid decline upon the addition of 0.3%. The reason is that CPAM exhibits enhanced bridging and net capture effects, resulting in the aggregation of small particles into clusters during the initial stage. This phenomenon shields a portion of the negative charge and temporarily decelerates the decline in zeta potential [38]. However, as more dispersant is added, the increased amount of negatively charged SHMP leads to a reduction in the zeta potential on the slurry surface.

The zeta potential only characterizes the stability of the macroscopic slurry system. To further investigate the influence of charge interactions from a microscopic perspective, we conducted a quantitative analysis of the dispersant’s impact on the effectiveness of flocculants in conjunction with DLVO theory. According to DLVO theory, the total potential energy V between particles is composed of the electrostatic interaction energy $V_{E}$ and the van der Waals interaction energy $V_{W}$, as depicted in Equation (1) [39].
(1)$$V=V_{E}+V_{W}$$

For particles of size $r_{1}$ and $r_{2}$, the van der Waals interaction energy $V_{W}$ between particles is as follows:(2)$$V_{W}=-\frac{A}{6H}\cdot \frac{r_{1}r_{2}}{r_{1}+r_{2}}$$

In Equation (2), H is the distance between particles. A is the effective Hamaker constant for the interaction between particle 1 and particle 2 in the medium, which is determined by Equation (3) below. $A_{11}$, $A_{22}$, and $A_{33}$ represent the Hamaker constants of particle 1, particle 2, and the medium in vacuum, respectively. As the main components of Zhanjiang clay are montmorillonite and kaolin, $A_{11}$ = 1.3 × 10^−19^ J, $A_{22}$ = 1.5 × 10^−19^ J, and $A_{33}$ = 4.4 × 10^−20^ J, based on ref. [40].
(3)$$A_{132}(A)=(\sqrt{A_{11}}-\sqrt{A_{33}})(\sqrt{A_{22}}-\sqrt{A_{33}})$$

The electrostatic interaction energy $V_{E}$ between particles is determined by Equation (4). $\varsigma_{1}$ and $\varsigma_{2}$ are the surface potentials of particles 1 and 2, respectively, which are approximately equal to their zeta potentials. For the relevant values, refer to Figure 19. ε is the dielectric constant of the medium. Due to the small dose of the reagent added, it is approximately considered equal to the dielectric constant ε of water = 7.11 × 10^−10^ F/m. κ is the Debye constant, which is determined by the ion concentration and valence of the flocculant. Here, κ = 1.432 × 109 m^−1^ is taken from ref. [41]:(4)$$\begin{array}{c} V_{E}=\frac{\pi \varepsilon r_{1}r_{2}}{(r_{1}+r_{2)}}(\varsigma_{1}{}^{2}+\varsigma_{2}{}^{2})\times \left[\frac{2\varsigma_{1}\varsigma_{2}}{\left(\varsigma_{1}{}^{2}+\varsigma_{2}{}^{2}\right)}x+y\right]\\ x=\ln\left[\frac{1+\exp(-\kappa H)}{1-\exp(-\kappa H)}\right],y=\ln\left[1-\exp(-2\kappa H)\right] \end{array}$$

To explore the essential relationship and simplify the calculation, it is assumed that the particle size of adjacent slurry particles is equal, that is, r = $r_{1}$ = $r_{2}$. For particle size per unit length:(5)$$\begin{array}{c} {\frac{V}{r}}_{}=\frac{\pi \varepsilon }{2}(\varsigma_{1}{}^{2}+\varsigma_{2}{}^{2})\times \left[\frac{2\varsigma_{1}\varsigma_{2}}{\left(\varsigma_{1}{}^{2}+\varsigma_{2}{}^{2}\right)}x+y\right]-\frac{A}{12H}\\ x=\ln\left[\frac{1+\exp(-\kappa H)}{1-\exp(-\kappa H)}\right],y=\ln\left[1-\exp(-2\kappa H)\right] \end{array}$$

Equation (5) and the aforementioned calculations were employed, and Figure 20 depicts the variation in the total potential energy per unit diameter between slurry particles as a function of particle spacing under the influence of two distinct flocculants with varying mass fractions of dispersants. The total potential energy curve of the intergranular interaction between the two flocculants exhibits an initial rapid increase followed by a gradual decline. The highest point on this curve, referred to as the “energy barrier,” represents its peak $V_{\max}$/r. The energy barrier is an energy index utilized for quantifying the magnitude of repulsion or attraction between slurry particles; particle energy must exceed this energy threshold via thermal motion and other mechanisms to undergo flocculation and precipitation [42]. Therefore, the lower the energy barrier is, the greater the enhancement in the flocculation and sedimentation capacity of slurry particles. Figure 20a demonstrates the impact of the dispersant on the energy barrier of the inorganic flocculant. With the addition of dispersants at concentrations of 0.1%, 0.3%, 0.5%, and 1%, the energy barrier between slurry particles resulted in distances of 2.7 nm, 2.5 nm, 2.2 nm, and 2.0 nm, corresponding to energy values of 101.5 × 10^−9^ J, 123.3 × 10^−9^ J, 162.2 × 10^−9^ J, and 191.7 × 10^−9^ J, respectively. The absence of dispersant results in an energy barrier of only 77.2 × 10^−9^ J; thus, the introduction of any concentration of dispersant can significantly increase the energy barrier between particles and subsequently enhance the repulsive force among slurry particles. This is also the direct reason why both high and low concentrations of the dispersant SHMP (Section 4.2.1) inhibited flocculation by inorganic flocculants, confirming that charge neutralization is the primary mechanism driving flocculation by the inorganic flocculant PAC. Figure 20b demonstrates the impact of the dispersant on the energy barrier of the organic flocculant. With the addition of dispersants at concentrations of 0.1%, 0.3%, 0.5%, and 1%, the energy barrier between slurry particles resulted in distances of 4.7 nm, 4.5 nm, 3.9 nm, and 3.5 nm, corresponding to energy values of 111.2 × 10^−9^ J, 117.1 × 10^−9^ J, 162.9 × 10^−9^ J, and 179.7 × 10^−9^ J, respectively. The dispersant exhibited a marginal increase in energy barrier at low concentrations (0.1% and 0.3%), with an average increase of only 8.5% compared to the baseline value of 105.2 × 10^−9^ J, indicating minimal change. The energy barrier of dispersants at high concentrations (0.5% and 1%) exhibited an average increase of 62.3%. The results presented in Section 4.3.2 demonstrate the effective enhancement in organic flocculant-induced bridging and net capture by dispersants, while this Section reveals that dispersants can slightly impede the charge neutralization action of organic flocculants at low concentrations. Despite these two effects, the superior flocculation performance exhibited by the organic flocculant indicates that CPAM’s primary mechanism for flocculation is bridging and net capture.

Cations present in the slurry are adsorbed onto the surfaces with negatively charged particles, forming a stable double electric layer [43]. After the addition of a flocculant to the slurry, there will be changes in both the type and concentration of ions present, as well as alterations in the thickness of the double electric layer due to electrostatic effects. The thinner the double electric layer is, the lower the energy barrier between particles, and the greater the likelihood of slurry flocculation and precipitation, as depicted in Figure 21a. However, the introduction of the negatively charged dispersant SHMP (as shown in Figure 21b,c) thickens the double electric layer of slurry particles and increases the energy barrier value between them. This affects the flocculant’s ability to carry out flocculation and is also why the dispersant impacts the charge-neutralizing effect of the flocculant. The analysis of energy barrier values above reveals that the dispersant SHMP exerts a more pronounced effect on the charge neutralization effect of the inorganic flocculant PAC than the organic flocculant CPAM, and this is primarily attributed to the distinctive molecular structures of the organic flocculants (Figure 21c).

#### 4.3.4. Effect of Dispersant on Vacuum Filtration

From the perspective of bridging and net capture effects, as well as charge neutralization, the trends and mechanisms governing the dispersant’s impact on slurry flocculation and precipitation effects were analyzed. Subsequently, by employing viscosity theory, we investigated the trends and mechanisms underlying the influence of dispersants on slurry vacuum filtration. The trend in viscosity in the slurry after the addition of the dispersant SHMP is illustrated in Figure 22 (the instrument shows a shear rate of about 20 s^−1^). The initial viscosity of the slurry was approximately 400 mPa.s. The initial slurry viscosity can be attributed to the negatively charged electric potential of slurry particles. Oxygen atoms on the surface of silicate mineral crystals, possessing a negative potential, attract hydrogen atoms with positive electric potential from water molecules, resulting in the formation of a highly viscous layer composed of water molecules [44]. As viscosity increases, so do the connections and adsorption between slurry particles, leading to an increased macrolevel viscosity in the slurry.

The viscosity of the slurry remained relatively unchanged after the addition of the inorganic flocculant PAC, whereas it exhibited a significant increase of approximately 37.5% upon the introduction of the organic flocculant CPAM. This disparity can be attributed to the bridging and net capture action of CPAM, which hinders the complete development of polymer chains within particles and consequently leads to entanglement with slurry particles, impeding their mutual movement in space [45]. Slurry viscosity with both inorganic and organic flocculants exhibited an initial decrease followed by an increase upon the addition of the dispersant SHMP. At a concentration of 0.1%, the viscosity of the organic flocculant reached its minimum level (353.3 mPa.s), while for the inorganic flocculant, it reached its lowest point (246.3 mPa.s) at a concentration of 0.3%. The decrease in slurry viscosity is attributed to the negative charge carried by SHMP molecules, which can effectively neutralize and shield the positively charged hydrogen atoms on the surface of water molecules in the slurry. This reduces the formation of highly viscous molecular water layers and indirectly enhances the liberation of bound water between slurry particles. As slurry viscosity decreases, the internal friction resulting from the relative movement or flow between molecules diminishes, leading to an increased degree of lubrication. Consequently, under negative pressure from vacuum filtration, the water in the slurry can be discharged more rapidly through interparticle seepage channels. The decrease in slurry viscosity leads to a reduction in internal friction caused by the relative movement or flow between slurry molecules, resulting in an increase in the degree of lubrication. This facilitates faster discharge of water from the slurry through the seepage channels present between particles under negative pressure during vacuum filtration. The viscosity of organic flocculants exhibited a rapid increase upon the addition of 0.1% dispersant, resulting in a significant elevation of slurry viscosity of 30.7 mPa.s for an average increment of 0.1% SHMP. Similarly, the introduction of 0.3% dispersant led to a substantial rise in the viscosity of inorganic flocculants, with an average increase of 0.1% SHMP and a total enhancement in slurry viscosity of 42.1 mPa.s. The reason for this may be attributed to the limited reactivity of excess SHMP within the slurry. Although SHMP is not a polymer per se, it undergoes decomposition and interpolymerization in aqueous solution, resulting in the formation of elongated chains akin to polymers. These elongated chains of SHMP intertwine with flocculant molecules, augmenting the viscous resistance of the slurry system. The increase in the viscous resistance of the slurry compared to untreated slurry not only obstructs the drainage channels between slurry particles but also decreases drainage efficiency, thereby impeding the settling of small slurry flocs. This directly accounts for improved vacuum filtration in the presence of low-concentration dispersants, as shown in Section 4.3.2 and Section 4.3.3, while high-concentration dispersants hinder the effectiveness of vacuum filtration.

In conclusion, the efficacy of organic or inorganic flocculants varies significantly depending on the mass fractions of the dispersant SHMP used. Therefore, precise control over the dose of added dispersant is essential during mud cake treatment in field engineering applications. This approach can be used to optimize flocculant performance in slurry separation and effectively enhance shield slurry separation efficiency while managing project costs.

## 5. Conclusions

In the context of the Zhanjiang Bay undersea Tunnel project in Guangdong Province, we conducted optimization tests for flocculants, tested the influence of dispersants on flocculation effects, investigated the mechanism by which dispersants affect flocculant performance, and explored both the effectiveness of flocculants and how dispersants influence flocculate efficacy. The objective of this article is to enhance the referencing and inspiration flocculants in the domain of slurry shields, thereby augmenting the efficiency and safety of slurry separation. The main conclusions are as follows:(1)Flocculation–sedimentation tests were employed to evaluate the effect of five different types of flocculants. Among them, PAC from the inorganic flocculant category exhibited superior performance, achieving optimal efficacy at a mass fraction of 0.5% when combined with slurry. CPAM from the organic flocculant category exhibited superior performance, achieving optimal efficacy at a mass fraction of 0.1% when combined with slurry.(2)The flocculation and sedimentation effects of the slurry are influenced by bridging and net capture effects and charge neutralization. Following the application of the organic flocculant CPAM and inorganic flocculant PAC, particles larger than 50 μm in the slurry accounted for 43.69% and 26.88% of particles, respectively. CPAM resulted in a higher accumulation of large particles, resulting in stronger bridging net formation. Additionally, upon the application of both the organic flocculant CPAM and inorganic flocculant PAC, the zeta potential of slurry particles increased to −21.9 mV and −12.6 mV, respectively; however, PAC resulted in more rapid attainment of strong flocculation potential due to its superior charge neutralization action.(3)Inorganic flocculants are inhibited by any fraction of SHMP in slurry flocculation. This is because while SHMP has little effect on PAC bridging and net capture, it significantly weakens charge neutralization, resulting in an energy barrier increase reaching 114.5 × 10^−9^ J at 1% addition. In the case of organic flocculants, a low concentration of SHMP (0.1% and 0.3%) promotes slurry flocculation, while a high concentration of SHMP (0.5% and 1%) strongly inhibits it. This phenomenon can be attributed to the fact that at lower concentrations, SHMP enhances bridging and net capture, resulting in an increase of 5.88% in the fraction of particles larger than 50 μm. However, at higher concentrations, SHMP primarily weakens charge neutralization, inhibiting overall slurry flocculation.(4)A low mass fraction of SHMP facilitates the vacuum filtration of slurry, while a high mass fraction of SHMP hinders it. This phenomenon can be attributed to the fact that a low mass fraction of SHMP reduces the viscosity of a slurry (reaching a minimum value of 246.3 mPa.s), decreasing internal friction and enabling faster discharge during vacuum filtration. Conversely, a high mass fraction of SHMP increases the viscosity of the slurry (up to 667.2 mPa.s), resulting in excessive entanglement caused by elongated chains formed by SHMP macromolecules, which obstructs drainage channels and decreases drainage efficiency.

## Figures and Tables

**Figure 1 polymers-15-04073-f001:**
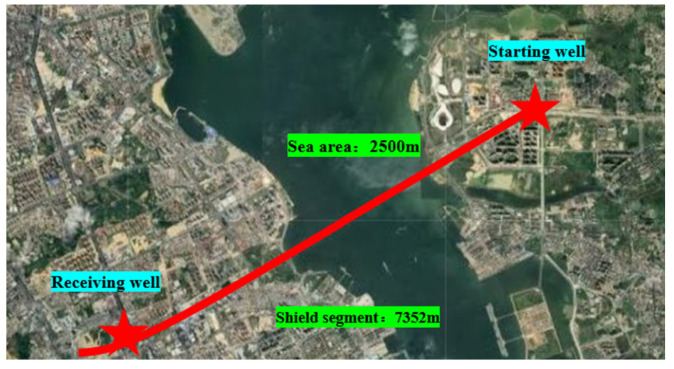
Zhanjiang Bay undersea tunnel.

**Figure 2 polymers-15-04073-f002:**
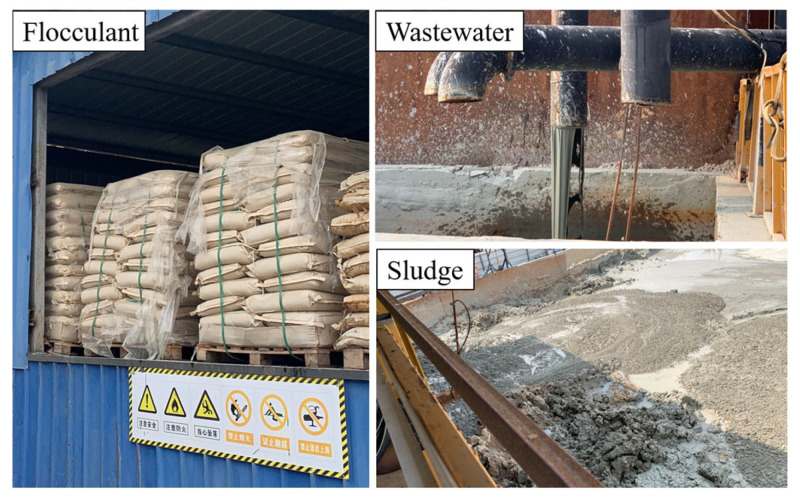
Flocculants, wastewater, and sludge.

**Figure 3 polymers-15-04073-f003:**
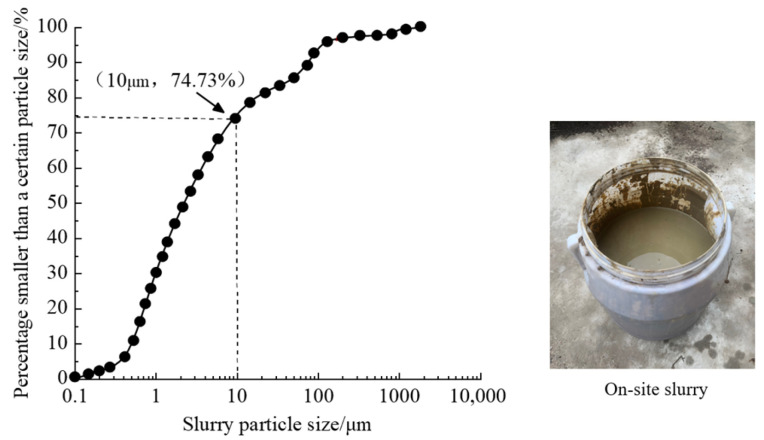
Particle distribution curve of slurry.

**Figure 4 polymers-15-04073-f004:**
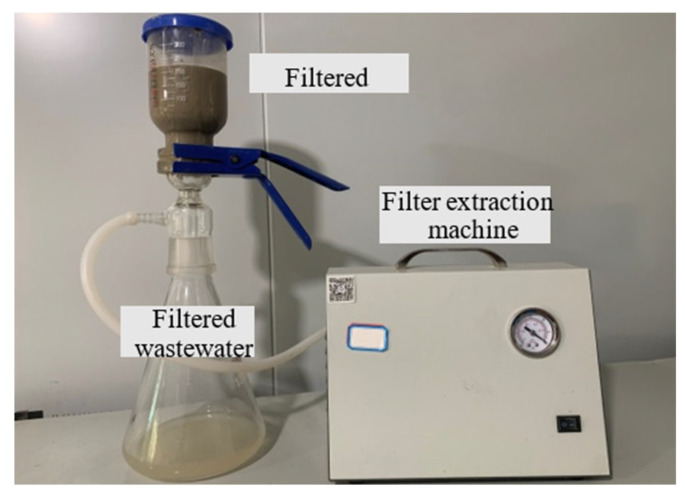
Vacuum-filtration device.

**Figure 5 polymers-15-04073-f005:**
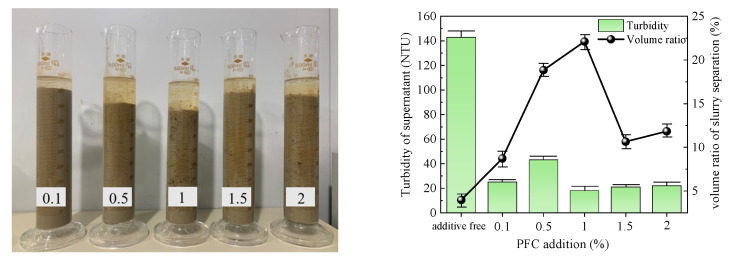
PFC test results.

**Figure 6 polymers-15-04073-f006:**
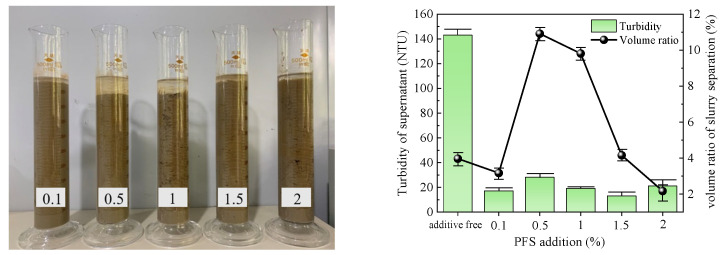
PFS test results.

**Figure 7 polymers-15-04073-f007:**
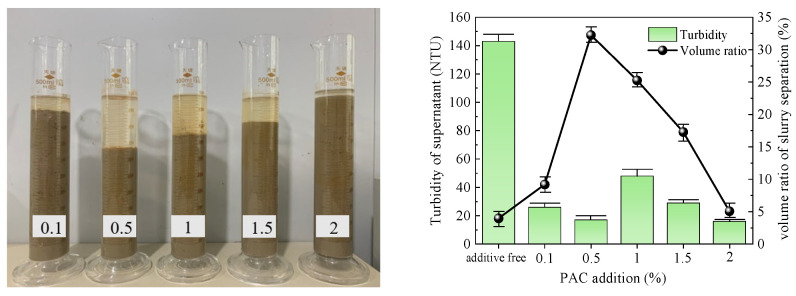
PAC test results.

**Figure 8 polymers-15-04073-f008:**
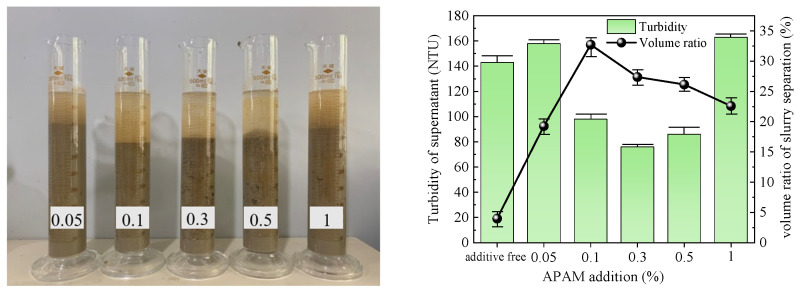
APAM test results.

**Figure 9 polymers-15-04073-f009:**
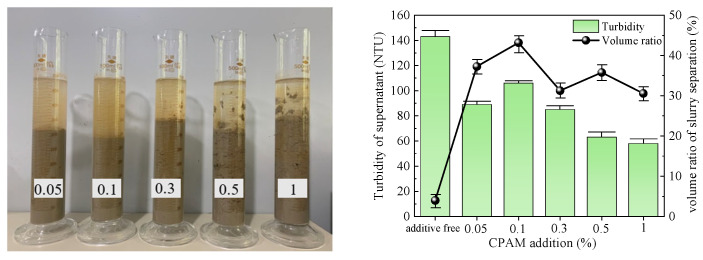
CPAM test results.

**Figure 10 polymers-15-04073-f010:**
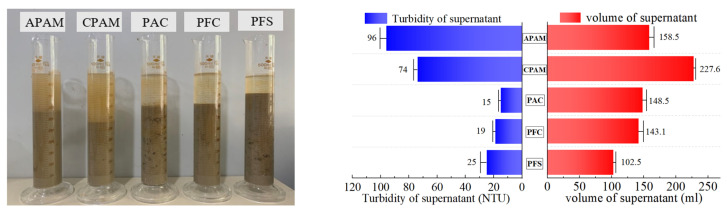
Result of flocculant screening test.

**Figure 11 polymers-15-04073-f011:**
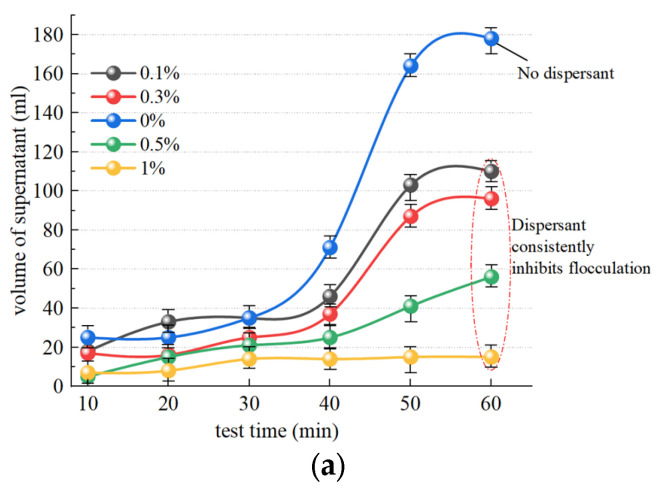
Effect of dispersants on the effect of inorganic flocculants. (**a**) Influence of SHMP on volume of supernatant; (**b**) influence of SHMP on turbidity of supernatant.

**Figure 12 polymers-15-04073-f012:**
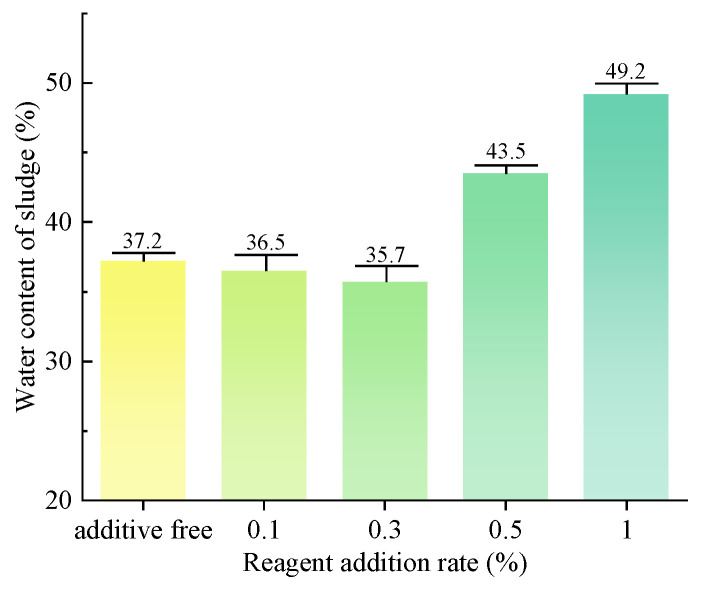
Water content of sludge after vacuum filtration. (group of inorganic flocculant).

**Figure 13 polymers-15-04073-f013:**
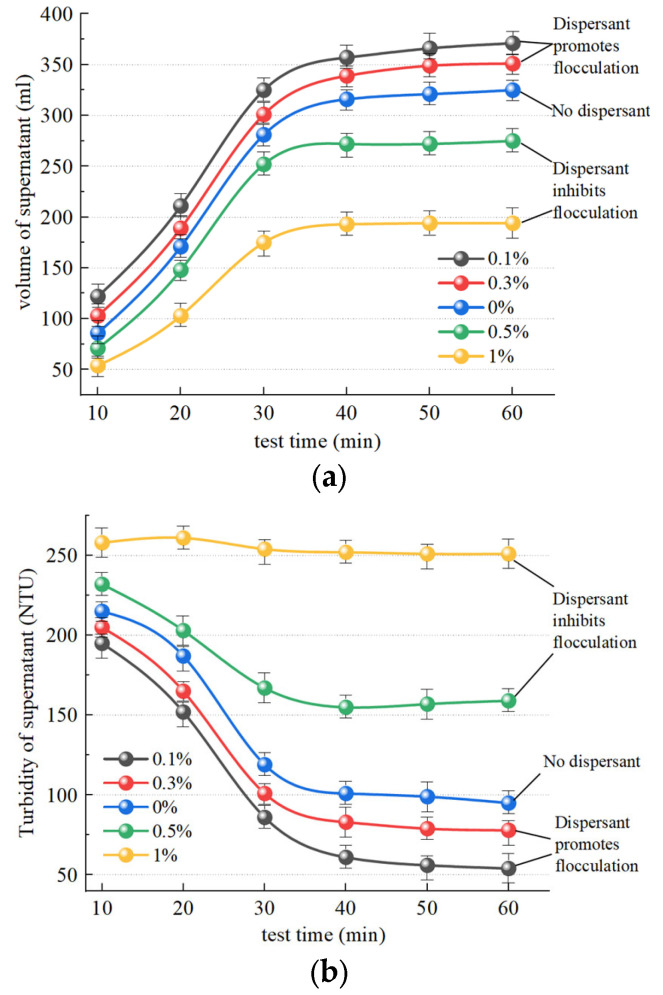
Effect of dispersants on the effect of organic flocculants. (**a**) Influence of SHMP on volume of supernatant; (**b**) influence of SHMP on turbidity of supernatant.

**Figure 14 polymers-15-04073-f014:**
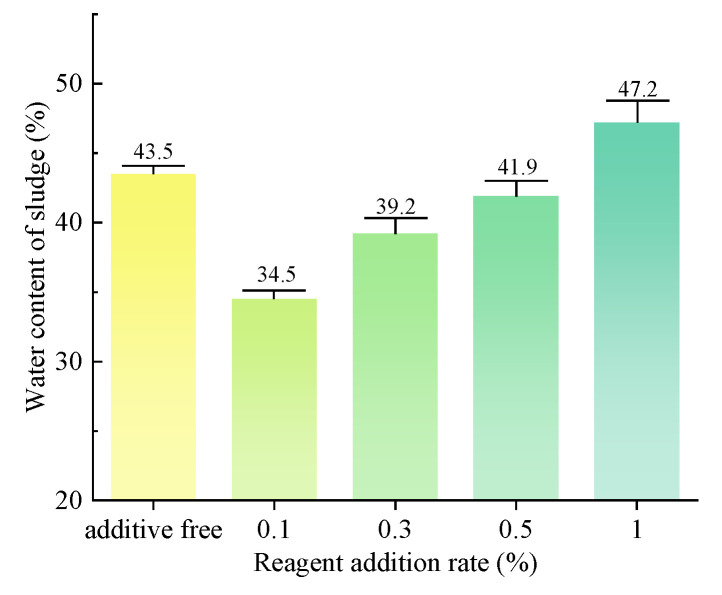
Water content of sludge after vacuum filtration. (group of organic flocculant).

**Figure 15 polymers-15-04073-f015:**
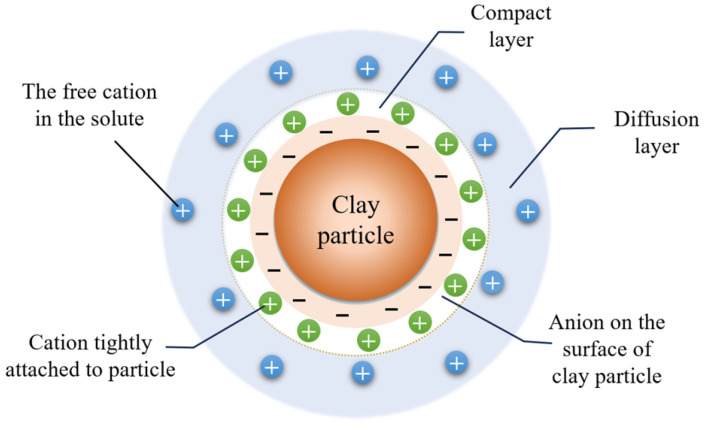
Double electric layer model.

**Figure 16 polymers-15-04073-f016:**
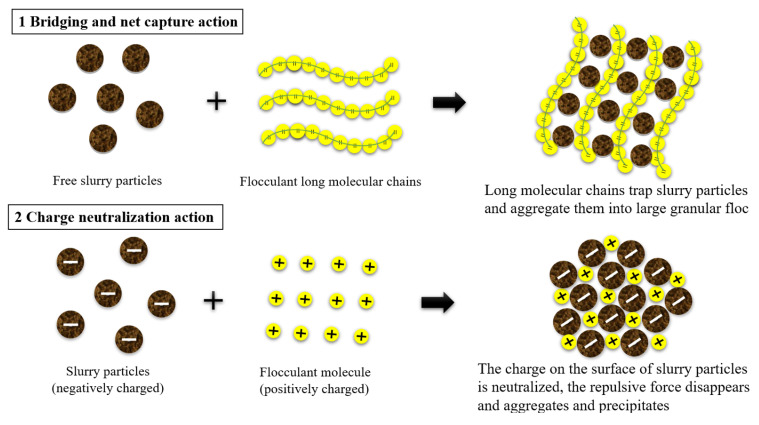
Flocculation mechanism of flocculants.

**Figure 17 polymers-15-04073-f017:**
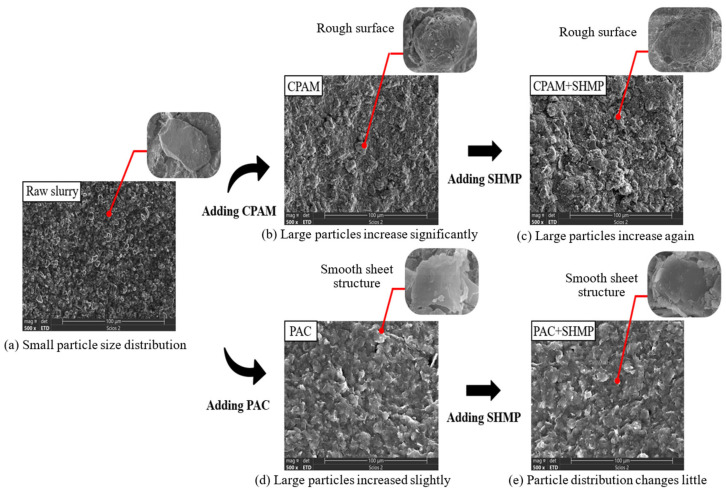
SEM images of slurry treated with different reagents.

**Figure 18 polymers-15-04073-f018:**
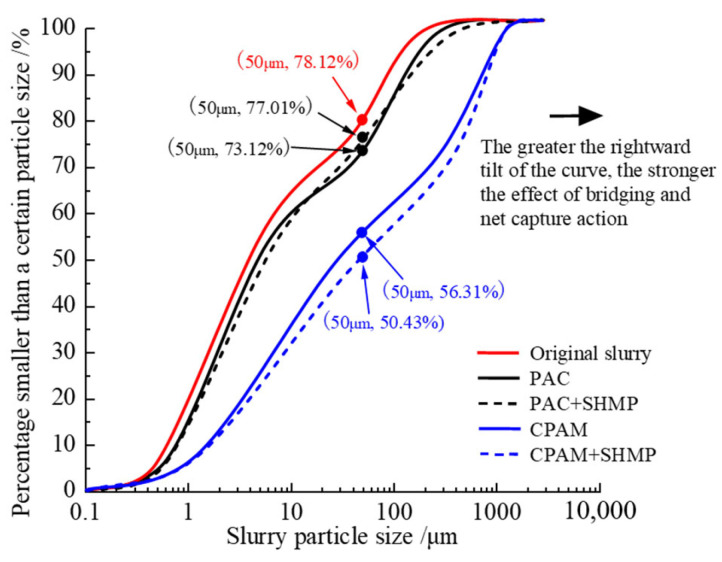
Particle size distribution of the slurry.

**Figure 19 polymers-15-04073-f019:**
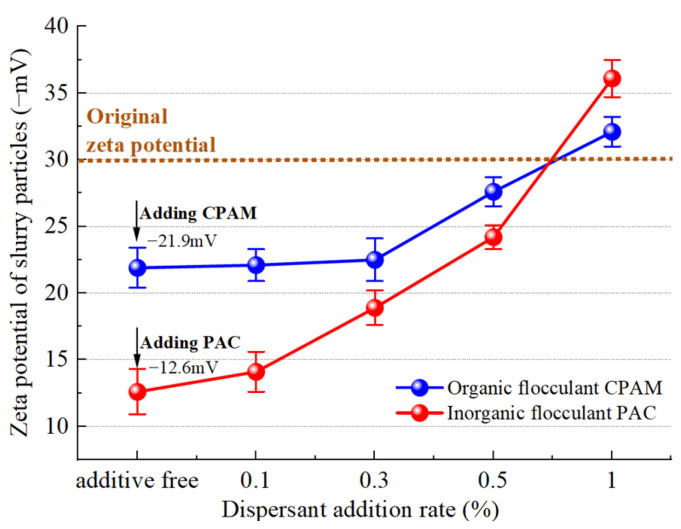
Variation trend of the potential.

**Figure 20 polymers-15-04073-f020:**
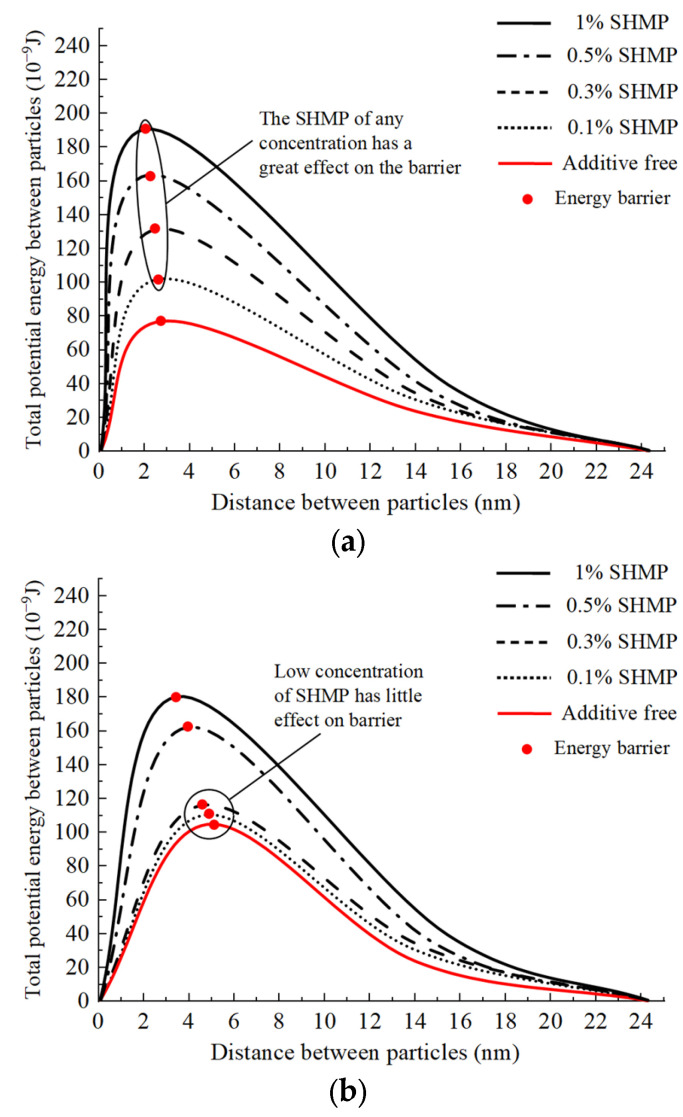
Changing trend of energy barrier. (**a**) Effect of dispersant on barrier of inorganic flocculant; (**b**) effect of dispersant on barrier of organic flocculant.

**Figure 21 polymers-15-04073-f021:**
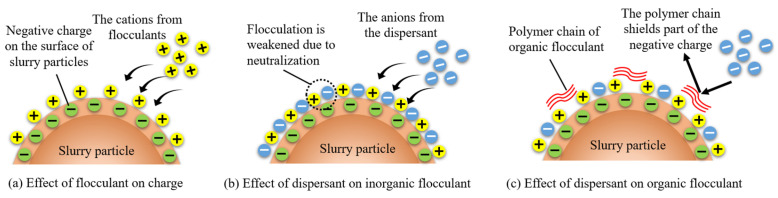
Action mechanism of different dispersants.

**Figure 22 polymers-15-04073-f022:**
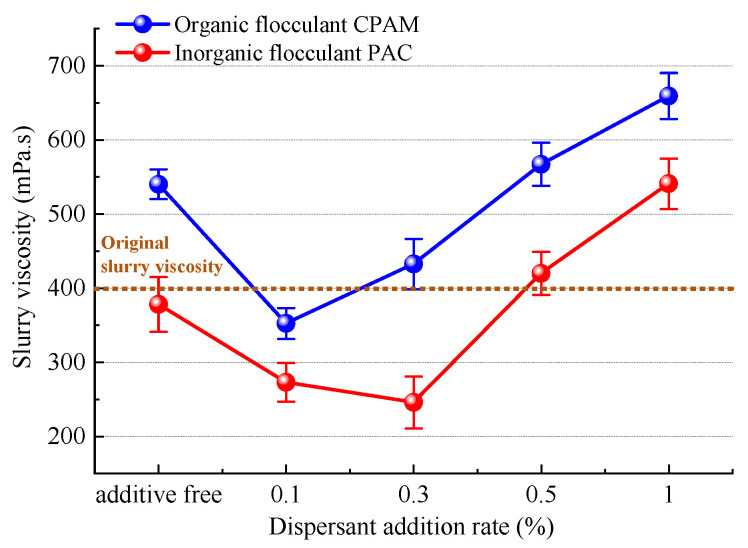
The viscosity trend of the slurry.

**Table 1 polymers-15-04073-t001:** Description of reagents. Reagents source: Guangzhou Lusheng Environmental protection Technology Co., LTD, Guangzhou, China.

Species	Reagent	Abbreviation	Properties
	Polyferric chloride	PFC	[Fe_2_(OH)_n_Cl_6−n_]_m_, reddish brown powder, highly soluble in water
Inorganic polymer flocculant	Polyferric sulfate	PFS	[Fe_2_(OH)_n_(SO_4_)_3−n/2_]_m_, pale yellow amorphous powder solid, soluble in water
	Polyaluminum chloride	PAC	[Al_2_(OH)_n_Cl_6−n_]_m_, yellow or light yellow, dark brown, dark gray resinous solid
Organic polymer flocculant	Anionic polyacrylamide	APAM	Polymers, white particles, high viscosity in aqueous solution, used mainly for flocculating positively charged colloid
	Cationic polyacrylamide	CPAM	Polymers, white particles, high viscosity in aqueous solution, used mainly for flocculating negatively charged colloid
Dispersant	Sodium hexametaphosphate	SHMP	(NaPO_3_)_6_, colorless transparent glass sheet or white granular crystal, easily delixed in air

**Table 2 polymers-15-04073-t002:** Description of the mechanism test.

Analysis	Instrument	Test Significance
	Hitachi Regulus 8100 Scanning Electron Microscope (Hitachi, Chiyoda City, Japan)	The mechanism by which a dispersant affects flocculation was analyzed from the perspective of apparent particle characteristics
Zeta potential	Malvern Zetasizer Nano ZS90 Potentiometric analyzer (Malvern Panalytical, Malvern, UK)	The mechanism by which a dispersant affects flocculation was analyzed from the perspective of potential
Granularity	Mastersizer 2000 Laser particle size analyzer (Mastersizer, Malvern, UK)	Combined with SEM images, the particle size distribution of clay particles under the action of different dispersants was analyzed
Viscosity	MDJ-9S Rotary viscometer (XiuiLAB, Shanghai, China)	The influence of viscosity on the effect of vacuum filtration was analyzed

**Table 3 polymers-15-04073-t003:** Summary of optimal additive filtering results.

Reagent	Group	Addition/%	Volume Ratio/%	Turbidity/NTU
PFC	1	0.1	8.72	25
	2	0.5	18.84	43
	3	1	22.12	18
	4	1.5	10.64	21
	5	2	11.83	22
PFS	6	0.1	3.16	17
	7	0.5	10.91	28
	8	1	9.81	19
	9	1.5	4.15	13
	10	2	2.16	21
PAC	11	0.1	9.18	26
	12	0.5	32.26	16
	13	1	25.26	48
	14	1.5	17.28	29
	15	2	5.05	17
APAM	16	0.05	19.26	158
	17	0.1	32.73	98
	18	0.3	27.39	76
	19	0.5	26.15	86
	20	1	22.58	163
CPAM	21	0.05	37.26	89
	22	0.1	43.18	106
	23	0.3	31.26	85
	24	0.5	35.78	63
	25	1	30.56	58

## Data Availability

Data will be made available on request.

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
