# Peer review of "Mechanism and Influence of Dispersants on the Action of Polymer Flocculants Used in Slurry Separation"

_polymers, 2023, doi:10.3390/polym15204073_

Round 1

Reviewer 1 Report

The manuscript reports the effect of flocculant/coagulant. While I cannot find scientific novelty in polymer science, some of data might be practically useful for engineers working on flocculation-sedimentation in civil engineering field. In this context, I am not sure if this manuscript is suitable for “Polymers”. The authors should emphasize the novelty and/or importance of this study for the readers of this journal.

All the powers should be properly written as superscripts. In the present manuscript, for example, “109, ten to the power of 9” is shown like “109”.

The authors focus on the effect of dispersant on flocculation. Such motivation was found in some studies such as a study on flocculation of ultrafine cement suspension by poly-γ-glutamic acid flocculant.

Is there a possibility of “sweep coagulation” mechanism for the authors’ systems as discussed for imogolite coagulant and aluminum salt?

The unit of interaction potential energy in Fig. 19 is strangely “1/m”. This unit should be “J”. Something is wrong.

Viscosity is slurry usually non-Newtonian. The authors should specify the value of shear rate when measuring the slurry viscosity.

Zeta potential is a value for particles. “zeta potential of slurry” sound strange.

Reviewer 2 Report

The manuscript was well organized. Knowledge gaps and the necessity of the work were indicated in the introduction. The main goal in the work was to find the optimum flocculant and the optimum conditions. There are major comments as follows:

1. Section 3.1, Figure 3: What methods were used to measure the particle size? DLS? Are volume or intensity-based sizes reported?

2. Figures 5, 6, 7, 8, 9, and 10: Error bars for all data must be included. The turbidity and separation of the control test (additive free) must be included.

3. Figures 11, 12, 13, 14: Error bars must be included.

4. Sections 4.2.1 and 4.2.2 are too long and must be made more concise.

5. Section 4.3.1 is very important. More relevant papers are required to support the mechanism. In addition, XDLVO theory (VDW, EDL, steric, and polymer bridging) must be included in the mechanism. In addition to scientific papers, I suggest including the book Food Emulsions: Principles, Practices, and Techniques by McClements because it concisely explains aggregation and flocculation mechanisms.

6. Section 4.3.3: The terms barrier potential should be replaced by energy barrier.

Reviewer 3 Report

The authors systematically studied the behavior of dispersants on the action of polymer flocculants used in slurry separation (inorganic and organic flocculants including other parameters like water content and dispersant were investigated). There are some comments to improve the quality of the paper:

   1- In lines 55-57, it is beneficial to add a brief summary of the impact of different types of flocculants on the efficiency of slurry separation (references 7 to 10). Only the title is mentioned in the introduction.

   2- In Fig 3, it is helpful to add a description of the method used to determine the particle size distribution of the slurry.

   3- The caption for Fig. 7 is missing. Also, the place of the graph versus the image should be revised. 

    4- Section 4.1.1, for different reagents (Figs 5-8) what is (are) the reason(s) for increasing and decreasing values of volume ratio of slurry separation and turbidity versus reagent content?

Round 2

Reviewer 1 Report

Again, the point  ”All the powers should be properly written as superscripts. In the present manuscript, for example, “109, ten to the power of 9” is shown like “109” is not modified.

Reviewer 2 Report

My comments have been addressed. I do not have comments for this revision and the manuscript is recommended for publication.

Author Response

Thank you for your comments. The quality of our paper has been enhanced thanks to your valuable comments and suggestions. We sincerely appreciate your contribution.